# Evidence for Establishing the Clinical Breakpoint of Cefquinome against Haemophilus Parasuis in China

**DOI:** 10.3390/pathogens10020105

**Published:** 2021-01-22

**Authors:** Kun Mi, Da Sun, Mei Li, Haihong Hao, Kaixiang Zhou, Zhenli Liu, Zonghui Yuan, Lingli Huang

**Affiliations:** 1National Reference Laboratory of Veterinary Drug Residues (HZAU), Wuhan 430000, China; mikun@webmail.hzau.edu.cn (K.M.); sundazz@126.com (D.S.); haohaihong@mail.hzau.edu.cn (H.H.); liuzhli009@mail.hzau.edu.cn (Z.L.); yuan5802@mail.hzau.edu.cn (Z.Y.); 2MAO Key Laboratory for Detection of Veterinary Drug Residues, Huazhong Agricultural University, Wuhan 430000, China; hzaulimei@163.com (M.L.); flyingkai@webmail.hzau.edu.cn (K.Z.); 3MOA Laboratory for Risk Assessment of Quality and Safety of Livestock and Poultry Products, Huazhong Agricultural University, Wuhan 430000, China

**Keywords:** clinical breakpoint, epidemiological cutoff, PK/PD cutoff, clinical cutoff, *Haemophilus parasuis*, cefquinome

## Abstract

*Haemophilus parasuis* can cause high morbidity and mortality in swine. Cefquinome possesses excellent antibacterial activity against pathogens causing diseases of the respiratory tract. This study aimed to establish the clinical breakpoint (CBP) of cefquinome against *H. parasuis* and to monitor the resistance change. Referring to the minimum inhibitory concentration (MIC) distribution of cefquinome against 131 *H. parasuis* isolates, the MIC_50_ and MIC_90_ were determined to be 0.125 and 1 μg/mL, respectively. And the epidemiological cutoff (ECOFF) value was 1 μg/mL. *HPS42* was selected as a representative strain for the pharmacodynamic (PD) experiment, pharmacokinetic (PK) experiment and clinical experiments. The PK/PD index values, area under concentration-time curve (AUC)/MIC, of the bacteriostatic, bactericidal, and bacterial elimination effects were 23, 41, and 51 h, respectively. The PK/PD cutoff was calculated as 0.125 μg/mL by Monte Carlo simulation (MCS), and the clinical cutoff was 0.25−4 μg/mL by WindoW. Combing these three values, the CBP of cefquinome against *H. parasuis* was found to be 1 μg/mL. In conclusion, this was the first study to integrate various cutoffs to establish the CBP in the laboratory. It is helpful to distinguish wild type *H. parasuis* and reduce the probability of treatment failure.

## 1. Introduction

Extensive agricultural use of antibiotics poses a risk of increasing antimicrobial resistance [1], which has been one of the main public health burdens. For controlling and monitoring the emergence of isolates with reduced susceptibility to antimicrobials, the clinical breakpoints (CBPs) are required to be set [2,3].

CBPs are used to categorize the results of antibiotic susceptibility testing (AST) as susceptible, intermediate, or resistant [4]. The European Committee on AST (EUCAST) and Clinical and Laboratory Standards Institute (CLSI) have proposed three cut-off values to constitute CBPs as follows: (1) CO_WT_, an epidemiological cut-off value (also known as ECOFF or ECV) which can differentiate the wild type (WT) from the non-wild strains; (2) Pharmacokinetic/Pharmacodynamic cutoff (PK/PD_CO_), a minimum inhibitory concentration (MIC) defined as PK/PD cut-off which is the highest MIC value for the most probable critical value (90%) in the target population with the calculated PK/PD index, such as AUC/MIC and T > MIC, where AUC is area under plasma concentration-time curve; and (3) CO_CL_, a MIC value related to the clinical therapeutic outcomes which could result in a high likelihood of successful therapy. Integrated with ECOFF, PK/PD_CO_, and CO_CL_, CBP could be ultimately determined by the characteristic procedures used in various organizations. VetCAST (Veterinary Antimicrobial Susceptibility Testing Subcommittee of EUCAST) and CLSI/VAST (Veterinary Antimicrobial Susceptibility Testing Subcommittee of CLSI) contributed to updating standardized CBPs processes and providing guidance and cases for researchers in the veterinary field. 

*Haemophilus parasuis* is a commensal gram-negative bacterium of the upper respiratory tract. It can invade the body in particular disease conditions. It can induce Glässer’s disease in piglets characterized by fibrinous polyserositis, arthritis, and meningitis [5]. *H. parasuis* can interact with other viruses, such as porcine reproductive respiratory syndrome virus (PRRSv), to infect piglets, resulting in high morbidity and mortality and a great financial loss [6]. There are 15 serotypes and many non-typeable serotypes of *H. parasuis* and they play a variant role in virulence [7]. For the diversity of *H. parasuis* serovars, vaccines only provide partial protection. Therefore, antimicrobial therapy is the primary method to treat *H. parasuis* diseases [8].

Cefquinome (CEQ), a fourth-generation aminothiazole cephalosporin, is approved by the European Medicines Agency for the treatment of swine respiratory tract disease at a dose of 2 mg/kg [9]. Multiple reports have suggested that cefquinome exhibits high efficacy against pathogens present in the swine respiratory tract, such as *Actinobacillus pleuropneumoniae* (*A. pleuropneumoniae*) [10], *Streptococcus suis* (*S. suis*) [11], and *H. parasuis* [12]. Because of the advantages of its structure, cefquinome conforms outstanding stability to the beta-lactamase and exerts excellent antibacterial activity against gram-positive and gram-negative bacteria in vitro or in vivo antibacterial activity [13] The PK of cefquinome has been characterized by rapid absorption and elimination and limited distribution in healthy piglets [14]. With these characterizations, cefquinome could be used for the treatment of *H. parasuis* infection. To our knowledge, no study has established the accurate CBP of cefquinome against *H. parasuis.* Since the ways to denote the antimicrobial susceptibilities of *H. parasuis* are based upon the breakpoints against *A. pleuropneumoniae* according to CLSI, they inevitably lead to an error and even undermine the potent effect of cefquinome in the veterinary field [15]. It is necessary to establish the CBP to monitor the trend of resistance of *H. parasuis*. 

Currently, various studies follow the guidelines of VetCAST and CLSI/VAST to estimate the breakpoints or ECOFF and PK/PD cutoff in veterinary applications. Toutain et al. adapted NLME and Monte Carlo simulation (MCS) to determine a MIC of 1 μg/mL as the PK/PD cutoff for the extensively used long-acting formulation of florfenicol for bovine respiratory disease (BRD) [16]. The result was similar to that of Lei et al., who derived the PK/PD cutoff of 1 μg/mL for florfenicol against *S. suis* in swine [17]. Additionally, Xiao has proposed a MIC of 0.06 μg/mL as the PK/PD cutoff for cefquinome against *H. parasuis* and analyzed other cutoffs [12]. The PK/PD cutoff could provide only some reference and gist, however, the accurate CBP need to combine the wild and clinical cutoffs. Schwarz summarized the available data for amoxicillin concerning PD, PK, clinical efficacy, and susceptibility to pathogens and proposed a CBP of amoxicillin against pathogens of the swine respiratory tract as follows: MIC <0.5 μg/mL, “susceptible”; MIC = 1 μg/mL, “intermediate”; and other MIC values, “resistant” [18]. CLSI has approved the CBP of florfenicol for BRD to be 2, 4, and 8 μg/mL, respectively, for “susceptible”, “intermediate” and “resistant” types [19]. However, CO_CL_ needs strict clinical conditions as a large number of animals are involved and the disease is only caused by the target pathogen. Therefore, few studies have investigated CO_CL_ under laboratory conditions and it was difficult to derive the accurate CBP without CO_CL_. Recently, a new method to derive the CO_CL_ has been proposed by Turindge [20], aiming to solve the dilemma to determine the CO_CL_ of CEQ against *H. parasuis*. 

In this study, we determined the MIC distribution of 131 *H. parasuis* isolates and integrated this with the results of a previous study to derive an ECOFF value. Then, we investigated the effect of cefquinome against *H. parasuis* using an ex vivo PK/PD model to derive the CO_PD_ by MCS. Next, we evaluated the clinical therapeutic outcomes of CEQ against *H. parasuis* disease from various MICs and adapted a novel algorithm to estimate the CO_CL_. Finally, we attempted to establish the CBP of cefquinome against *H. parasuis* for monitoring the changing trend of resistance. 

## 2. Results

### 2.1. MIC Distribution of CEQ against H. parasuis and ECOFF Calculation 

The MIC of 131 *H. parasuis* ranged from 0.0075 to 8 μg/mL, as can be seen in Figure 1A. The MIC_50_ and MIC_90_ values were calculated as 0.125 μg/mL and 1 μg/mL, respectively. The wild-type strains were statistically discriminated by the ECOFFinder based on Turnidge. Four different endpoints (95%, 97.5%, 99%, and 99.5%) were calculated as 4, 4, 8, and 16 μg/mL. Generally, the ECOFF should be set at least encompassing 95% wild-type strains and the ECOFF value of CEQ against *H. parasuis* was 4 μg/mL in this study. Previously, Xiao investigated the MIC distribution of cefquinome against 213 isolates of *H. parasuis*. We combined the results to obtain an expansive MIC distribution (Figure 1B). The MIC distribution of cefquinome against 344 *H. parasuis* was imputed into the software and the 95%, 97.5%, 99%, and 99.5% endpoints were calculated as 1, 2, 4, and 4 μg/mL, respectively. Conclusively, the ECOFF of CEQ against *H. parasuis* was 1 μg/mL. 

### 2.2. Selection of Strains and MIC and MBC 

Eight *H. parasuis* strains (serotype 5) were selected from MIC_90_ to evaluate the virulence by mice experiments (as Appendix A shown). Because *HPS42* possessed the strongest virulence with the obvious diseased symptoms, it was selected for the PD experiment. The MIC and MBC of cefquinome against *HPS42* in TSB were 1 and 2 μg/mL and in serum were 0.5 and 1 μg/mL. The ratio of MBC/MIC was 2, which signified that cefquinome might have a strong bacteriostatic activity both in vitro and ex vivo. Furthermore, based on the broth: serum MIC ratios (1 μg/mL:0.5 μg/mL), the antibacterial effect of cefquinome against *H. parasuis* is similar in different mediums, which reveals no serum effect on the potency of cefquinome [21,22]. 

### 2.3. Time–Killing Curves 

The time–killing curves of cefquinome against *HPS42* in vitro and ex vivo are shown in Figure 2 and Figure 3. The curves showed a time-dependent antibiotic activity and the optimal bactericidal activity showed a threshold of approximately 4 × MIC. A further increase in the concentration resulted in a similar efficacy [23,24]. 

### 2.4. PK of Cefquinome

The pigs in the infected group did not manifest any adverse reactions before *H. parasuis* incubation. After 2 days of consecutive bacterial challenge, the pigs showed symptoms of fever and depression at 48 h. The concentration–time profiles of cefquinome in the serum of healthy and diseased pigs are shown in Figure 4 following administration of i.m. cefquinome (2 mg/kg). 

The PK parameters of cefquinome in healthy and *H. parasuis* infected pigs, derived from non-compartmental analysis, are shown in Table 1. The PK parameters of cefquinome in the serum of healthy and diseased pigs were calculated by WinNonlin. Cefquinome was rapidly absorbed within 0.25 h from the injection site and reached the C_max_. The C_max_ in the diseased pig was twice that in the healthy group. The AUC, T_1/2_, and MRT of cefquinome in the healthy and diseased pig sera were 8.61 ± 2.68 and 15.52 ± 4.07 h × μg/mL; 3.52 ± 0.81 and 3.19 ± 0.92 h; and 4.61 ± 0.68 and 3.59 ± 0.88 h, respectively.

### 2.5. PK/PD Integration and Analysis 

The PK/PD parameters were determined from the integration with in vivo PK data and the ex vivo MIC values. The ratio of C_max_/MIC, AUC_24_/MIC, and T>MIC were 5.05, 17.21 h, and 5.25 h in healthy pigs, whereas these values were 11.49, 31.04 h, and 7.34 h in diseased pigs, respectively. The PK/PD parameters and AUC_24_/MIC fitted into the inhibitory sigmoid E_max_ model. Table 2 indicates the model parameters and various antibacterial effects. 

### 2.6. PK/PD Cutoff Calculation of Cefquinome against HPS42

With the bactericidal effect, the target endpoint for the PK/PD index (AUC_24 h_/MIC) in serum is 41 h. The probability of target attainment (PTA) was calculated by different MIC values. The PTA was 90.41% at a concentration of 0.125 μg/mL with an increase in the MIC of cefquinome against *H. parasuis*, and it gradually declined to 0% at a concentration of 0.5 μg/mL. Consequently, the CO_PD_ of cefquinome against *H. parasuis* was calculated to be 0.125 μg/mL. 

### 2.7. Determination of the CO_CL_

One isolate *H. parasuis* should be selected from MIC_50_, MIC_90_, PK/PD_co_, ECOFF, and the highest MIC of the test population to infect swine, respectively. Due to PK/PD_co_ = MIC_50_ = 0.125 μg/mL, MIC_90_ = ECOFF = 1 μg/mL, the sensitive strain (MIC = 0.25 μg/mL, *HPS64*) and the resistant strain (MIC = 4 μg/mL, *HPS47*) are selected for the clinical cutoff experiment. In addition, PK/PD_co_ = MIC_50_ = 0.125 μg/mL (*HPS80*), MIC_90_ = ECOFF = 1 μg/mL (*HPS42*), and the highest MIC = 8 μg/mL (*HPSL23*) were also selected. By WindoW, the CO_CL_ is calculated objectively by a mathematical method. For treatment, each group was administered with intramuscular cefquinome (2 mg/kg). The WindoW approach consists of two separate algorithms, MaxDiff and CAR, to reduce the influence of subjectivity. Prior to the use of WindoW, some rules need to be followed. The CAR value cannot be set at the boundary and the estimation could be operated unless the experiment isolates are > 4 (n > 4), etc. 

According to previous research [20], the MaxDiff value was calculated as 16.67 which corresponds to a MIC of 4 μg/mL, as well as the cumulative success rate (CAR) value 0.5 for a MIC of 0.125 μg/mL, although it could not be set as the boundary. Therefore, the CO_CL_ was defined from 0.25 to 4 μg/mL. The result of WindoW is shown in Table 3.

### 2.8. Establishment of CBP 

The CBP of cefquinome against *H. parasuis* needs to refer to three values. Because of an ECOFF of 1 μg/mL, PK/PD_CO_ of 0.125 μg/mL, and CO_CL_ of 0.25−4 μg/mL, we suggested setting the CBP of cefquinome against *H. parasuis* at 1 μg/mL.

## 3. Discussion

*H. parasuis*, the main pathogen causing respiratory tract diseases, is a serious threat to the survival of weaner pigs. With the use of antibiotics, the resistance of *H. parasuis* to antibiotics has gradually increased [25,26]. Cefquinome possesses excellent antibacterial activity for the initial treatment of *H. parasuis* infection. Therefore, to monitor the change in resistance and reduce the failure ratio of clinical treatment, establishing the CBP of cefquinome against *H. parasuis* is a top priority. 

In China, the antibacterial susceptibility of cefquinome against *H. parasuis* has not changed much from 2015 to 2017. Xiao et al. determined the MIC of cefquinome against 213 isolates of *H. parasuis* and a MIC_50_ of 0.125 μg/mL was found, which is similar to the current result; however, fewer strains were tested in this study, and the MIC_90_ of 8 μg/mL varied from a MIC_90_ of 1 μg/mL [12]. In Brogden’s study, a MIC_50_ of ≤ 0.015 μg/mL and a MIC_90_ of 0.06 μg/mL of cefquinome against 123 isolates of *H. parasuis* were reported from Germany [27]. The reason for the different antimicrobial susceptibility results was because of the geographical differences that could be frequently found [28,29]. Meanwhile, the MIC determination method could also result in discrimination. As the method approved by CLSI for the determination of the MIC of *H. parasuis* was unavailable, Brogden adopted the microdilution broth method by CAMHB and the others used the agar dilution method by tryptose soya agar. In addition, the typical ECOFF needs more data, using the exact same method from multiple labs.

ECOFF does not correspond to the clinical therapeutic outcome but it can be a predictor of pathogen resistance to antibacterial when no breakpoint is established [30]. If the MIC derived from the antibacterial susceptibility test is above the ECOFF, the isolates will be non-wild, in which the clinician would consider administering other drugs. Numerous methods have been proposed to determine the ECOFF [31,32,33,34]. Meanwhile, Turinidge’s method [34] and Kronvall’s method [32] are universally recognized. In Kronvall’s method, normalized resistance interpretation (NRI) is used to define the WT population in the inhibition zone diameter histograms [35,36]. An algorithm is available on the website (http://www.bioscand.se/nri/). For minimum inhibitory concentration, ECOFFinder has been used to calculate the epidemiological cutoff value by Turinidge’s method, which could facilitate the computational process [37,38].

It has been proposed that EUCAST and CLSI use a minimum of 100 isolates for each bacterial species to derive the ECOFF, and additionally, at least 30 WT needs to be included to derive the ECOFF [16,39,40]. Few studies have reported the epidemiological cut-off values in veterinary and the ECOFF of tilmicosin against *H. parasuis* has been set at 16 μg/mL. Amoxicillin, as a widely used beta-lactam antibiotic, was determined the ECOFF to *P. multocida* and *A. pleuropneumoniae* in swine as 1 or 0.5 μg/mL by the visual method, which showed a reduced susceptibility as compared to Garch’s method [41,42]. In our study, we determined the MIC distribution of cefquinome against 131 isolates of *H. parasuis* and derived the ECOFF as 4 μg/mL. Furthermore, we combined our results with the published results and considered the geographical position and culture medium, by running the ECOFFinder; the ECOFF was determined to be 1 μg/mL.

A total of 131 *H. parasuis* were isolated and identified by PCR [43]. *HPS42* is the most virulent strain with a MIC_90_ of 1 μg/mL of cefquinome against *H. parasuis* and was selected for further study. In some previous studies, the target strain of the tested population was irregularly selected and *SH0165* is a popular selection that ignores the characterization of *H. parasuis* in the wild type [17,44]. In contrast, the most representative strain can be selected on a reliable basis by our method [38]. If cefquinome can effectively inhibit the most virulent strain from MIC_90_ of the population, the estimation of the PK/PD model will be much more valuable.

The first step to estimate the PK/PD_CO_ is to determine the PK/PD index derived from the PK/PD model. The AUC/MIC, C_max_/MIC, and T>MIC were empirically used as the PK/PD index [45]. However, the variable values of the T>MIC vs. bacterial count could not be directly obtained from the ex vivo PK–PD model [14]. In this study, we used an inhibitory sigmoidal E_max_ model, and the PK/PD index (AUC/MIC) showed a favorable correlation (R^2^ = 0.9926) with the predicted antibacterial efficacy. This indicated AUC/MIC could be the optimal PK/PD index for PK/PD integration model. Some published articles have employed AUC/MIC as a PK/PD index for cefquinome [14,24]. Florfenicol is classified as a time dependent drug; however, AUC/MIC has often been proposed as an optimal PK/PD index that is predictive of clinical efficacy [17,46]. A semi-mechanistic PK/PD model of florfenicol against *P. multocida* and *M. haemolytica* was applied in in silico simulations to predict AUC/MIC, and outperformed T> MIC as the PK/PD index [47]. Due to its ethical and economic advantages, a semi-mechanistic PK/PD model could have a wider application in veterinary science [4,16].

In the next step, by Monte-Carlo simulation, the conservative value (AUC/MIC = 41 h) was selected to calculate the PTA. When the MIC was 0.125 μg/mL and the PTA was 90.41%, the value was reduced to <90% by increasing the MIC. Xiao reported a PK/PD_CO_ value of 0.06 μg/mL of cefquinome against *H. parasuis* [12] which is similar to the current PK/PD_CO_ value of 0.125 μg/mL. Of note, the PK/PD_CO_ value (0.125 μg/mL) was less than the CO_WT_ value (1 μg/mL), which was probably due to unknown resistance mechanisms or a lower dose of drug administered to pigs. Moreover, a PK/PD_CO_ of 0.125 μg/mL of cefquinome against *H. parasuis* was much lower than those of tilmicosin (1 μg/mL) [44] and marbofloxacin (0.5 μg/mL) [48], implying more clinical potency of cefquinome than other antibacterials against *H. parasuis*.

Beta-lactam antibiotics, including penicillins, cephalosporins and carbapenems, are widely used in veterinary clinics. The production of beta-lactamase is one of the major resistance mechanisms in gram-negative bacteria. β-Lactam resistance in *H. parasuis* is related to plasmid pB1000, which bears the *blaROB-1* β-lactamase [49]. Moreover, biofilm has been associated with resistance to β-lactams in *H. parasuis* [50]. The MecA gene or the CTX-M gene is considered to positively influence the susceptibility of cefquinome against *staphylococcus aureus* and *escherichia coli*, respectively [51,52].

It is acceptable to integrate the results of ECOFF, PK/PD_CO_, and CO_CL_ to establish the CBP. Some difficulties hinder the establishment of CO_CL_; for instance, in practice it is very difficult to distinguish between curable and non-curable diseases solely based on the MIC and find adequate cases infected with the non-wild-type strains. Besides, many random factors affect treatment outcomes, and hence, it is hard to find the relationship between MIC and clinical outcomes for CO_CL_. The clinical cutoff, CO_CL_, can minimize the risk of treatment failures; however, no veterinary case has demonstrated the relationship between the MIC of antibiotics against the target pathogen and the cure rate [4]. Due to the absence of CO_CL_ by the standard method, we adopted a less commonly used method to determine the clinical cutoff under laboratory conditions. WindoW, a new approach described to calculate CO_CL_ by Turnidge [20], integrates two separate algorithms, MaxDiff and CAR, and recognizes the potential presence of microbial distributions which have clinical relevance. A CO_CL_ value of 0.25−4 μg/mL can be calculated by WindoW. For the first time, we used WindoW to establish CO_CL_ in the veterinary context in order to provide a reference. Combined with clinical outcomes, CBP is more representative and accurate in monitoring resistance.

The CBP involves comparison among ECOFF, PK/PD cutoff, and clinical cutoff. EUCAST proposed a method that only includes ECOFF and PK/PD breakpoint except for the clinical cutoff. Aside from integrating the conceptual framework of the PK/PD vs. clinical outcome relationship, they did not describe and use clinical cutoff [4]. Following the method to establish the breakpoint, if the PK/PD breakpoint is higher or equal to ECOFF, the PK/PD breakpoint could be selected as the CBP; otherwise, ECOFFs could be recommended. CLSI demands three cutoffs to establish the breakpoint, and remarkably, for the PK/PD cutoff, they used the PK data to meet the PK/PD targets [19]. Recently, Papich et al. followed the documents to determine the CBP of cephalexin against *Staphylococcus pseudintermedius* in dogs. Firstly, they determined the MIC distribution, and secondly, they used the PK data to ensure that the dosage was effective. Finally, by Monte-Carlo simulations, they derived the breakpoints of ≤2 µg/mL (susceptible), 4 µg/mL (intermediate), and ≥8 µg/mL (resistant) [53].

We derive ECOFF = 1µg/mL, PK/PDco = 0.125 µg/mL and CO_CL_ = 0.25−4 μg/mL. However, there is no guideline to follow when the CO_CL_ is not an exactly MIC value. VetCAST recommends ECOFF as a surrogate when the CBP is not established [4]. As Lei reports [38], while ECOFF > PK/PDco, ECOFF also plays an important role in establishing the CBP. For our study, ECOFF is within the range of CO_CL_ and PK/PDco is out of it. Considerable for the importance of ECOFF, we establish the CBP as 1 µg/mL. Compared to the published breakpoint, our formulated CBP is equal to ampicillin, ceftibuten, and doxycycline against *Haemophilus influenzae* in humans [54]. It is higher than amoxicillin (0.5 µg/mL) [18] and lower than tildipirosin (4 µg/mL) [38] for swine respiratory tract pathogens.

There are some limitations in our manuscript. We preliminarily proposed ECOFF, PK/PDco and CO_CL_, respectively. Based on the current results, the CBP of cefquinome against *H. parasuis* is advised to be 1 μg/mL. For an ultimate CBP, more results of the CBP and cutoffs from other institutions are needed. Furthermore, the results of clinical therapy experiments in the wild need to be gathered to narrow the range of CO_CL_. In our study, an ECOFF of 1 µg/mL, PK/PDco of 0.125 µg/mL, and CO_CL_ of 0.25−4 μg/mL were determined. We estimated the CBP of cefquinome against *H. parasuis* (1 μg/mL), which referred to ECOFF PK/PDco and CO_CL_. This result will supply some evidence for the further study of the establishment of clinical breakpoint in laboratory conditions.

## 4. Materials and Methods

### 4.1. Animals

Thirty-six six-week-old crossbred (Duroc × Large White × Landrace) pigs (weighing 15–20 kg) were purchased from the Livestock and Poultry Breeding Centre of Hubei Province (Wuhan, China). Before the experiment, all pigs were raised for 7 days to get acclimatized and were not allowed to take any antibiotics. The diseased model was followed from the study of Zhang where pigs were inoculated with 10^9^ CFU/mL of *H. parasuis* (1−2 mL) in each nostril four times on two consecutive days [55]. After experiment, cefquinome was administrated by intramuscular injection to diseased swine at a dose of 2mg/kg/24h for 3 days. Through etiological surveillance and clinical symptoms observation for seven days, healthy swine were returned to the farm. Diseased swine were euthanized. All efforts were used to reduce the pain and adverse effect of the animals.

Seventy-five six-week-old female BALB/c mice (specific pathogen free grand; bodyweight of 18 ± 2 g) were obtained from the Center of Experimental Animal of Hubei and housed in the SPF animal room in the laboratory. The research was approved by the Ethics Committee of the Faculty of Veterinary Medicine of the Huazhong Agricultural University. All animal experiments were conducted according to the committee guidelines for the Laboratory Animal Use and Care Committee in Hubei Science and Technology Agency. All efforts were used to reduce the pain and adverse effect of the animals. After the experiments, all mice were euthanized.

### 4.2. Strains and Antibiotic

A total of 131 *H. parasuis* strains stored at −80 °C in milk were used to determine the MIC of cefquinome by agar dilution (Dr. Ehrenstorfer Standards, Augsburg, Germany) at the National Reference Laboratory of Veterinary Drug Residues and State Key Laboratory for Detection of Veterinary Drug Residues at Huazhong Agricultural University. *A. pleuropneumoniae* (ATCC 27090) was used as the quality control strain (QC). Tryptone soya agar (TSA) and Tryptone soya broth (TSB), supplemented with 5% fetal calf serum and 1% nicotinamide adenine dinucleotide (NAD, 10 μg/mL), were used to culture *H. parasuis*.

### 4.3. MIC Distribution and ECOFF Determination 

The antibacterial susceptibility testing was performed by the agar dilution method based on CLSI [56]. *H. parasuis* were inoculated onto TSA plates containing cefquinome (0.0075−8 μg/mL) by serial twofold dilutions. The plates were incubated at 37 °C in an atmosphere containing 5% CO_2_ for 36 h. The MIC contained a minimum amount of cefquinome where the visible growth of bacteria was inhibited. The values of MIC_50_ and MIC_90_ in the test population were calculated by SPSS version 19.0.

The ECOFF value was described as the upper limit of the wild population which comprised 95% strains of the MIC distribution. A conventional statistical method has been proposed by Turnidge [56]. Briefly, the WT distribution was checked for normality by SigmaStat software, and nonlinear regression was performed to calculate the mean and standard deviation by GraphPad Prism. Finally, the NORMINV and NORDIST functions of Microsoft Excel were applied to set the ECOFF. Additionally, the ECOFFinder program, based on the above method, was used to simplify the statistical method and calculate the ECOFF. It can be found on the website (http://www.eucast.org/mic_distributions_and_ecoffs/).

### 4.4. Selection of the Virulent Strains

The Balb/c mice model was used as an alternative model to evaluate the virulence of *H. parasuis* in pigs [57]. Seventy-five female mice were divided into one control group (n = 3) and eight infection groups (n = 72) challenged by different *H. parasuis* (serovar 5) chosen from MIC_90_. Each infection group was intraperitoneally administrated for three infection doses (10^7^, 10^8^, and 10^9^ CFU/mL with 0.5 mL). Three mice were infected with each dose for an infection group. The control group was challenged by 0.5 mL of PBS. During 72 h after infection, the number of deaths among mice was monitored and the organs were observed after anatomy to evaluate the most virulent strain from MIC_90_ selecting for PD experiment.

### 4.5. Pharmacodynamics Experiments

#### 4.5.1. Determination of MIC and MBC

The MIC of cefquinome against *HPS42* was determined with serial two-fold dilution by broth microdilution technique following the guidelines of the CLSI at serial concentrations between 8 and 0.0075 μg/mL. For the MBC of cefquinome against *HPS42*, 100 µL of suspension from the 96 well plates of CEQ were diluted with TSB by 1:10 steps and 10 µL was spread on TSA agar plates for the colony-forming unit (CFU) counting and incubated at 37 °C with 5% CO_2_ for 24 h. The MBC was the minimum concentration of cefquinome inhibiting 99.9% of the bacterial density.

#### 4.5.2. Time–Killing Curves In Vitro and Ex Vivo 

The bacteria (10^6^ CFU/mL) were cultured with a two-fold dilution of cefquinome ranging from 1/4 to 16 × MIC. The growth was compared with the control. The tubes were incubated at 37 °C with 5% CO_2_ and the viable counts of bacteria were determined at 0, 1, 2, 3, 4, 6, 8, 10, 12, and 24 h. At each time point, 100 µL of aliquots were serially diluted by saline, and then, the CFUs were counted after 24 h of incubation. The limit of detection was 10 CFU/mL.

The plasma samples obtained from the healthy and diseased group was considered as a culture media for the ex vivo MIC, MBC, and time–killing curves. The bacteria (10^6^ CFU/mL) cultured with the plasma samples were collected at 0.083, 0.167, *0*.25, 0.5, 0.75, 1, 1.5, 2, 3, 4, 6, 8, 10, 12, and 24 h after cefquinome administration. The tubes were incubated at 37 °C with 5% CO_2_ and the viable counts of bacteria were determined at 0, 3, 6, 9, 12, and 24 h after co-culture.

### 4.6. Pharmacokinetic Experiments

#### 4.6.1. Sample

Blood samples were collected from healthy (n = 6) and diseased groups (n = 6). Each pig received cefquinome (cefquinome sulfate 2.5%, Amicogen (Jining, China) Biopharm Co., Ltd.) by intramuscular injection (single dose of 2 mg/kg). Blood samples were collected with an anticoagulant at 0.083, 0.167, 0.25, 0.5, 0.75, 1, 1.5, 2, 3, 4, 6, 8, 10, 12, and 24 h after administration. The samples were centrifuged at 3500 *× g* for 10 min and then stored at −20 °C prior to HPLC and PD experiment.

#### 4.6.2. HPLC

Waters 2695 series HPLC and a Waters 2587 UV detector set at a wavelength of 265 nm equipped with ZORBAX SB-Aq column (250 × 4.6 mm i.d., 5 µm; Agilent Technology, Santa Clara, CA, USA) were used. The injection volume was 30 µL and the temperature was maintained at 30 °C. The mobile phase consisted of A (0.1% phosphoric acid) and B (acetonitrile) with gradient elution as follows (0−8 min, 10% B, 8–8.1 min, 15% B; 8.1−15 min, 10% B) at a flow rate of 1 mL/min.

The sample was extracted with 1 mL of acetonitrile. The tubes were vortexed for 2 min and then centrifuged at 10,000× *g* for 10 min. Following that, 1.5 mL of dichloromethane was added, and the tubes were vortexed for 15 s and centrifuged at 10,000× *g* for 10 min. The supernatant was filtered through a 0.22 μm nylon Millipore chromatographic syringe filter for HPLC.

The method was validated with reference to the residue guidelines of the Veterinary Pharmacopoeia of the Department of Agriculture and the Pharmacopoeia of the United States (Gad, 2014). The validation of linearity, limit of determination (LOD), limit of quantitation (LOQ), accuracy, and precision of the method were determined by blank serum with CEQ standard solution. The calibration curves were constructed using blank serum with CEQ at six levels as follows: 0.05, 0.1, 0.1, 1, 5, and 10 μg/kg (n = 3). The calibration curve was y = 63065x − 3595.1, (R^2^ = 0.9996). The LOD value was 0.03 μg/mL which was the lowest detected concentration at the value of the signal to noise ratio (S/N) > 3 in serum. LOQ was 0.05 μg/mL, which was the lowest detected concentration at the value of S/N > 10 in serum. LOD and LOQ were defined three times and, respectively, established by the following steps: 15 blank serum added drugs were analyzed, and the S/N was calculated at the time window in which the analyte was expected. Accuracy and precisions (intraday, interday, and within laboratory) were calculated from the determination of five aliquots of serum at 0.1, 1, and 5 μg/kg. The recovery of CEQ in plasma ranged within 94−99%, with the intraday relative SD less than 13%. The PK data were analyzed with WinNonlin V5.2.1.

### 4.7. PK/PD Integration Analysis 

For the PK/PD model, AUC_24 h_/MIC, C_max_/MIC, and T>MIC are considered as the standardized PK/PD index. These parameters were determined by combining time–killing curves and the in vivo PK parameters. Using WinNonlin version 5.2.1, the inhibitory sigmoid *E_max_* model was used to evaluate the correlation of index in vitro and the change in the bacterial count following 24 h of incubation. The applicable model equation was described as Equation (1) [58].
(1)$$E=E_{max}-\frac{(E_{max}E_{0})\cdot C^{N}}{C^{N}+EC_{50}^{N}}$$
where, *E* is the PD endpoint, *E*_0_ is the change in log_10_ CFU/mL after 24 h incubation in the control sample as compared to the initial incubation, *E_max_* is the differential effect between the greatest amount of growth and the greatest amount of kill, *C* is the PK/PD index (AUC_24 h_/MIC, C_max_/MIC or T > MIC) in the compartment, *EC*_50_ is the PK/PD index value producing 50% reduction in the bacterial count from initial inoculum, and *N* is the Hill coefficient that describes the steepness of the curve.

### 4.8. Monte-Carlo Analysis and PK/PD Cutoff Calculation 

Monte-Carlo simulation (MCS) is a mathematical technique that relies upon repeated random sampling to evaluate the impact of uncertainty when characterizing the probability of an outcome [59]. For PK/PD_CO_, Crystal Ball software version 7.2.2 was used to perform Monte-Carlo simulation based on the PK/PD target index (AUC_24h_/MIC, E = −3, bactericidal activity). Simultaneously, it was calculated as the highest MIC for the probability target attainment exceeding 90% according to the CLSI guidelines and other previously described studies [34].

### 4.9. Clinical Cutoff

The conventional clinical cutoff method, the “eyeball approach”, is inadvertently influenced by subjective assessment. In the mathematical method, WindoW aims to reduce the subjective error associated with CO_CL_ assessment by identifying the inflection point of MIC with the rate of clinical therapeutic change. One strain (serotype 5) from MIC_50_, MIC_90_, CO_PD_, CO_WT_, and highest MIC of the test population should be selected. In the clinical experiment, 30 pigs were divided into five diseased groups (n = 6), which were infected by the above strains, and the control group (n = 6). The physiological parameters, including body temperature, mental state, and respiratory symptoms, were monitored to adjudicate the administration cefquinome. The cure rate of each group after administration of i.m. cefquinome (2 mg/kg once daily for 3 days) was noted to calculate CO_CL_ by WindoW.

## Figures and Tables

**Figure 1 pathogens-10-00105-f001:**
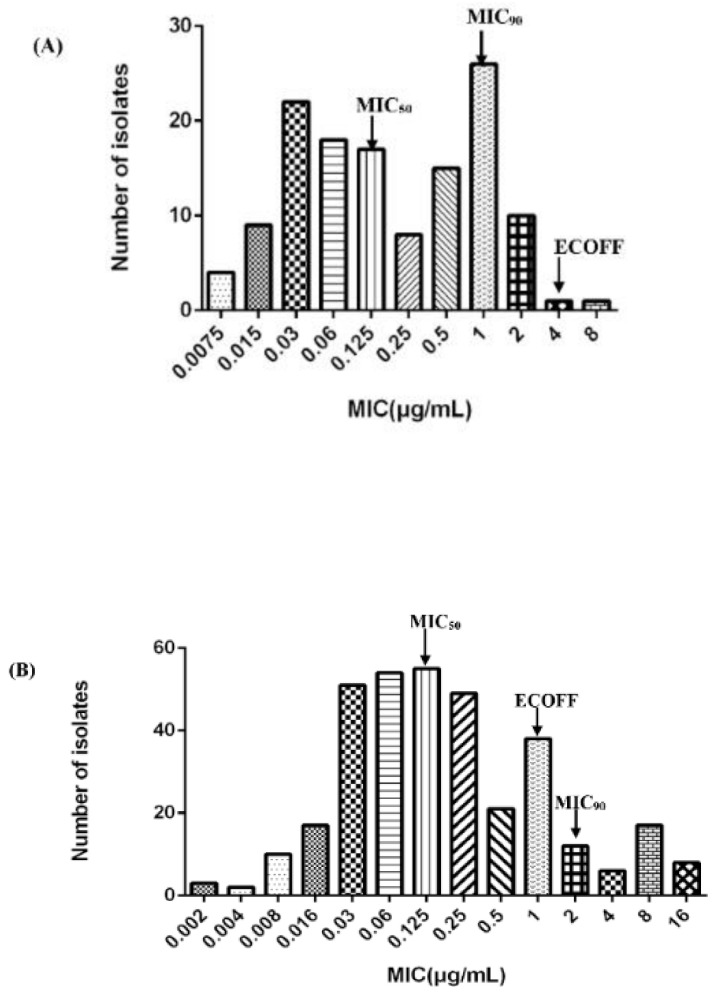
Distribution of Minimum Inhibitory Concentration (MIC) of cefquinome against *H. parasuis*. (**A**) The number of strains is 131; (**B**) combined with Xiao’s result [12], the number of strains is 344.

**Figure 2 pathogens-10-00105-f002:**
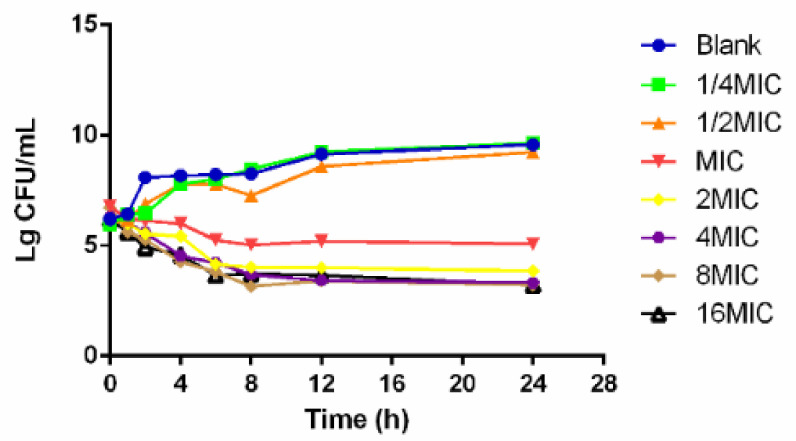
The in vitro time–killing curve of cefquinome against *HPS42*.

**Figure 3 pathogens-10-00105-f003:**
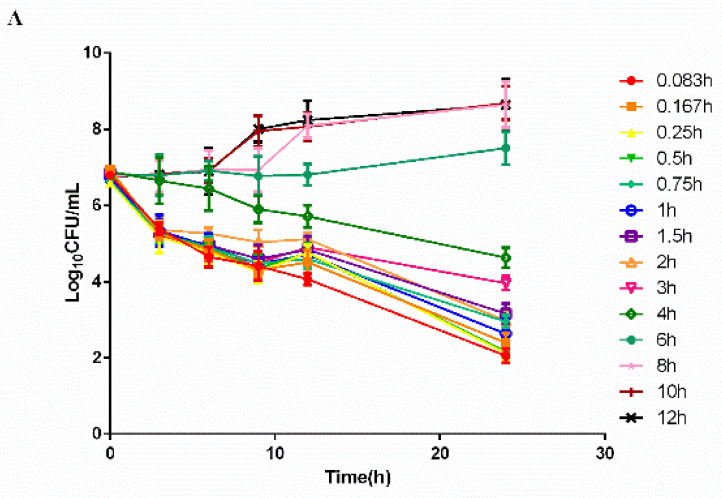
The ex vivo time–killing curves in serum. (**A**) Represented curves in healthy group and (**B**) Represented curves in diseased group. Note: Bacterial number was determined at different time points by a variety of serum samples from the pharmacokinetic (PK) study. Legends represents the cefquinome concentration in the sampling time point.

**Figure 4 pathogens-10-00105-f004:**
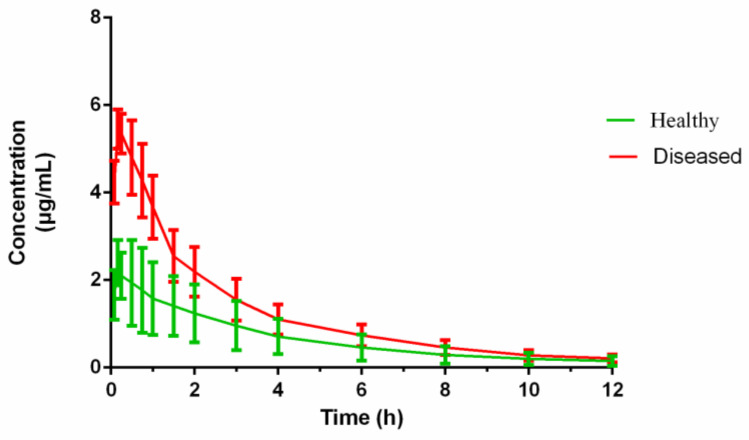
The concentration–time curves of cefquinome in serum from the healthy pigs (n = 6) and the diseased pigs (n = 6) after I.M. administration 2mg/kg bodyweight (b.w.).

**Table 1 pathogens-10-00105-t001:** Pharmacokinetic parameters of cefquinome after intramuscular (I.M.) administration (2 mg/kg) in healthy and *H. parasuis*-infected swine (n = 6).

Parameters	Unit	Healthy	Infected
T_1/2_	H	3.52 ± 0.81	3.19 ± 0.92
*T* _max_	H	0.24 ± 0.12	0.26 ± 0.11
*C* _max_	μg/mL	2.53 ± 0.46	5.75 ± 0.59
AUC _0–∞_	hr*μg/mL	8.61 ± 2.68	15.52 ± 4.07
Vz_F	mL/kg	1.39 ± 0.74	0.63 ± 0.27
Cl_F	mL/h/kg	0.26 ± 0.09	0.14 ± 0.03
MRT_last_	H	4.61 ± 0.68	3.59 ± 0.88

C_max_: maximal drug concentration; Tmax: time to reach C_max_; T1/2: the half-life; CL_F = clearance per fraction absorbed; Vz_F = volume of distribution per fraction absorbed; MRT = mean residence time.

**Table 2 pathogens-10-00105-t002:** The ex vivo Pharmacokinetic/Pharmacodynamic (PK/PD) parameters after I.M. administration cefquinome for various antibacterial effects.

Parameter	Unit	Healthy	Diseased
E_max_	Log10CFU/mL	1.71	2.10
EC_50_	Log10CFU/mL	33.20	47.66
E_0_	Log10CFU/mL	−4.06	−4.34
N	_	3.61	3.45
AUC_24h_/MIC for bacteriostatic (E = 0)	H	23	39
AUC_24h_/MIC for bactericidal (E = −3)	H	41	70
AUC_24h_/MIC for eradiction (E = −4)	H	51	110

CFU: count forming unit; EC50: the PK/PD index value producing 50% reduction in the bacterial count from initial inoculum.

**Table 3 pathogens-10-00105-t003:** The result of WindoW by different clinical treatment groups.

MIC (μg/mL)	Success	Failure	Total	%Success ≤MIC	%Success >MIC	MaxDiff	AUC Success	AUC Total	CAR
0.125	3	3	6	50.00	62.50	−12.50	0.19	0.38	0.50
0.25	4	2	6	58.33	66.67	−8.33	0.63	1.13	0.56
1	4	2	6	61.11	66.67	−5.56	3.63	5.63	0.64
4	5	1	6	66.67	50.00	16.67	17.13	23.63	0.72
8	3	3	6	63.33	100.00	−36.67	33.13	47.63	0.70

## Data Availability

The data presented in this study are available on request from the corresponding author.

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
