# Peer review of "Evidence for Establishing the Clinical Breakpoint of Cefquinome against Haemophilus Parasuis in China"

_pathogens, 2021, doi:10.3390/pathogens10020105_

Round 1

Reviewer 1 Report

This is a well written paper presenting well design research on Haemaphilus parasuis presenting the resistance of some Chinese strains to Cefquinome as a one of the most effective antibiotic in bacterial diseases treatment.

Author Response

Dear reviewer:  

  Thanks for your work. We will continue to revise our manuscript to reach the publishing requirements. 

Best regards.

Kun Mi

Reviewer 2 Report

This study aims to establish a veterinary clinical breakpoint of cefquinome for Haemophilus parasuis, a clinically important pathogen for pigs. The authors combine different approaches and integrate various cutoffs to establish the CBP in the laboratory, which is helpful to address their aims. The manuscript is well-written and the experiments are designed thoughtfully. 

-I believe the paper is suitable for publication in Pathogens, but there is a need to strengthen the discussion by covering the most important mechanism of resistance for cefquinome and their impact on MIC and their relative prevalence. 

-There is no explanation about the origin of the strains. To propose a CBP, isolates should have a representative geographical distribution. Is it the case in this study?

-Proposition of CBF also benefits from multicenter studies from different cities. However, this is a study from a few centres located within the same city. This limitation should be discussed in the manuscript.  

-The authors combine three cutoff values to propose the CBP of cefquinome to Haemophilus parasuis. How the authors equilibrate the contribution of each one? The authors should discuss better why this combination is better than a single cutoff. I could not find in the manuscript the rationale behind defining a CBP.  

-How can the authors make sure their strain collection is representative of the pathogen? Is there any strain typing available to conclude that they represent different strain and not many isolates from a few strains? If this information is not available, the authors should point that as a limitation of the study. 

-The figure legends are very poor. One has to go back to the text to find additional explanations. There are no references about what represent error bars and there is neither any statistical inference. Everything looks very qualitative over quantitative data. The statistical analysis can be certainly improved. In many cases, it is simply inexistent while the authors have quantitative information that is not analyzed.  

-Figure 4. What are the units in the Y-axis (Log10 concentration of what and in what?). Is the log scale needed in this graph? 

-Some abbreviations are not depicted. For example EUCAST, CLSI, and PPSRV 

-The authors say in the introduction: Since the ways to denote the antimicrobial susceptibilities of H. 67 parasuis is based upon the breakpoints against A. pleuropneumoniae according to CLSI. What is that cutoff value and how different it is from that one proposed by the authors?

Author Response

Dear reviewer:

Thanks for your work.Please see the attachment

Best regards.

Kun Mi

Reviewer 3 Report

Evidence for establishing the Clinical Breakpoint of Cefquinome against Haemophilus parasuis in China by Mi et al is reviewed.  This paper builds on work modeling PK/PD  and MICs for H. parasuis and cefquinome published in 2015 by adding clinical data and establishing a clinical breakpoint. 

General Comments:  The paper needs English revision throughout.  However, the overall approach is sound.  My strongest concerns regarding this are lack of detail in the results and methods.  It would be impossible to reproduce this data as written.  Details should be included in both materials/methods and the results sections to be able to thoroughly evaluate this manuscript. 

Specific Comments:

ECOFF values were calculated by combining multiple data sets to get to 344 isolates.  In the methods it is not clear how this was done.  Was it the same lab method? 

Line 108: Results are incomplete.  Please provide results and how the strains were evaluated in mouse model.

Figure 2 and 3: were these replicated experiments?  Are can bars be added to indicate distributions?

Figure 3 Legend.  Disease group of pigs?  Mice please edit legend

Line 126 Results are incomplete.  Please summarize clinical scores/findings, etc from pig study.

Discussion needs edited overall for length and clarity.  There also appear to be results in the discussion which should be moved to that section.

Line 279: Animal study methods are incomplete.  What kind of pigs/age/size weight?  Which strain were they inoculated with?  Were they euthanized or necropsied and how ere they sampled.  Need a lot more information here to evaluate the study.

Line 289-303: The manuscript indicates CLSI VET01-A4 for A. pleuropneunmoniae.  However the methods indicate that media was used with 5% calf serum and 1% NAD.  CLSI guidannce in Table 7 of Vet01-S2 indicates that chocolate MHA shouldbe used for agar diffusion.  Serum has been shown to potentiate the antimicrobial effect, at least for some drugs.  Was this tested if serum was used?  The methods also indicate that they were incubated for 36 hours and CLSI methods require 20-24 hours  Please clarify.  Were both of these combined data sets tested using the same methods? Please indicate any modifications to CLSI methods.  It would be inappropriate to combine the two data sets of MIC if different methods were used. 

I have similar questions about the MIC/MBC testing (321-327).  What broth was used?  Was it the serum+NAD broth?  Was this validated.  CLSI recommends the use of vet fastidious media and has recently approved MHF-Y for APP and H. somni.  Was this used or was serum media validated for this use?  Please describe the methods.  

Author Response

(The authors gave the same response as above.)

Round 2

Reviewer 3 Report

The revised manuscript Evidence for establishing the Clinical Breakpoint of Cefquinome against Haemophilus parasuis in China is reviewed.  The changes are much improved in addressing feedback.  There is still some language polishing needed in regards to singular/plural usage and verb tenses in some areas.  

Specific Comments:

For the ECOFF values on 344 isolates, suggest adding in discussion that typically ECOFF estimates require data from multiple labs using the exact same method.  This is just two labs, so in discussion need to indicate this requires more data. 

Line 108:  I still am not able to find in the results the data from the mice challenge experiment.  Please add this.  FIgure 1 in the response could be used as a supplemental figure.  How many mice/strain?  How many had lesions?  were they scored? Etc..

Line 126 Table 1 could added to supplemental or if there is an reference or this is a standard scoring could be added.  

Line 206.  "the antibacterial susceptibility test is above the ECOFF, the isolates will be wild, in which the clinician would consider administering other drugs."

Do the authors mean "non-wild" i.e. they have acquired resistance above background?  I am not understanding this. 

Line 257. The added text needs editing.  There need to be appropriate italics and verb tense.  

I agree that the broths/agars supplemented with NAD/FCS are appropriate here, however, there is some inconsistency. Line 328 indicates CLSI guidelines for APP were followed for the method.  However, the methods used by the authors are not this.  Suggestions would be "were based on CLSI" with the deviations (media, incubation times, etc) to the CLSI that were used.  

Author Response

Specific Comments: 

Q1 :For the ECOFF values on 344 isolates, suggest adding in discussion that typically ECOFF estimates require data from multiple labs using the exact same method.  This is just two labs, so in discussion need to indicate this requires more data.

Answer: Thanks for your suggestions. As you said, ECOFF of cefquinome against haemphilus parasuis need data from multiple labs using the exact same method. Line 202, We have added this part in the discussion..

Q2: Line 108:  I still am not able to find in the results the data from the mice challenge experiment.  Please add this.  FIgure 1 in the response could be used as a supplemental figure.  How many mice/strain?  How many had lesions?  were they scored? Etc..

Answer:  Thanks for your suggestion. We will add the data of mice challenge experiment in supplementary materials.

Q3: Line 126 Table 1 could added to supplemental or if there is an reference or this is a standard scoring could be added.    Line 206.  "the antibacterial susceptibility test is above the ECOFF, the isolates will be wild, in which the clinician would consider administering other drugs."  Do the authors mean "non-wild" i.e. they have acquired resistance above background?  I am not understanding this.

Answer:  Thanks for your suggestion. We will add the Table 1 in the supplemental. Line 206. Thanks for your reminder, we have revised the “wild” to “non-wild” in the manuscript.

Q4: Line 257. The added text needs editing.  There need to be appropriate italics and verb tense.    I agree that the broths/agars supplemented with NAD/FCS are appropriate here, however, there is some inconsistency. Line 328 indicates CLSI guidelines for APP were followed for the method.  However, the methods used by the authors are not this.  Suggestions would be "were based on CLSI" with the deviations (media, incubation times, etc) to the CLSI that were used. 

Answer:  Thanks for your suggestion. We have revised the ” added text” from line 249 to 254. And, line 324, we revised the method of determining MIC as your suggestion. Thank you again.